# Mapping Coastal Wetlands Using Satellite Imagery and Machine Learning in a Highly Urbanized Landscape

**Juan Munizaga** [1] , **Mariano García** [2] , **Fernando Ureta** [3] , **Vanessa Novoa** [1] , **Octavio Rojas** [1] and **Carolina Rojas** [4],*

1   Departamento de Planificación Territorial y Sistemas Urbanos, Facultad de Ciencias Ambientales y Centro EULA-Chile, Universidad de Concepción, Víctor Lamas 1290, Concepción 4070386, Chile; juanmunizaga@udec.cl (J.M.); vanessanovoa@udec.cl (V.N.); ocrojas@udec.cl (O.R.)
2   Universidad de Alcalá, Departamento de Geología, Geografía y Medio Ambiente, Environmental Remote Sensing Research Group, C/Colegios, 2, Alcalá de Henares, 28801 Madrid, Spain; mariano.garcia@uah.es
3   Departamento de Ingeniería Metalúrgica, Facultad de Ingeniería, Universidad de Concepción, Víctor Lamas 1290, Concepción 4070386, Chile; fernandoureta@udec.cl
4   Instituto de Estudios Urbanos y Territoriales, Pontificia Universidad Católica de Chile, Centro de Desarrollo Urbano Sustentable CEDEUS, Instituto Milenio en Socio-Ecología Costera SECOS, El Comendador 1916, Providencia, Santiago 7820245, Chile
*   Correspondence: carolina.rojas@uc.cl

**Abstract:** Coastal wetlands areas are heterogeneous, highly dynamic areas with complex interactions between terrestrial and marine ecosystems, making them essential for the biosphere and the development of human activities. Remote sensing offers a robust and cost-efficient mean to monitor coastal landscapes. In this paper, we evaluate the potential of using high resolution satellite imagery to classify land cover in a coastal area in Concepción, Chile, using a machine learning (ML) approach. Two machine learning algorithms, Support Vector Machine (SVM) and Random Forest (RF), were evaluated using four different scenarios: (I) using original spectral bands; (II) incorporating spectral indices; (III) adding texture metrics derived from the grey-level covariance co-occurrence matrix (GLCM); and (IV) including topographic variables derived from a digital terrain model. Both methods stand out for their excellent results, reaching an average overall accuracy of 88% for support vector machine and 90% for random forest. However, it is statistically shown that random forest performs better on this type of landscape. Furthermore, incorporating Digital Terrain Model (DTM)-derived metrics and texture measures was critical for the substantial improvement of SVM and RF. Although DTM did not increase the accuracy in SVM, this study makes a methodological contribution to the monitoring and mapping of water bodies' landscapes in coastal cities with weak governance and data scarcity in coastal management.

**Keywords:** coastal wetlands; remote sensing; coastal cities; RapidEye; machine learning

## 1. Introduction

Coastal wetland areas are ecotones or interfaces between terrestrial and marine ecosystems, where the lithosphere and hydrosphere interact with highly complex and dynamic mechanisms [1,2]. These areas are recognized for their importance to the biosphere [3,4]. In addition, among the most critical ecosystems are estuaries, mangroves, coral reefs, intertidal habitats and deltas, dunes and beaches, seagrass, kelp forest, marshes and swamps, water bodies, among other habitats [5,6].

These coastal ecosystems provide numerous fundamental services for the development of human activities, such as raw materials and food, protection against waves and storms, erosion control, carbon sequestration, tourism, recreation, and water purification [7]. Because of these multiple benefits, it is possible to find large concentrations of the human

population and socioeconomic activities around these areas. They host a third of the world's population [8,9], presenting a higher population density, growth, and urbanization than inland areas [10]. They have also contributed to generating transport links, economic activities, tourism, and industrial and commercial development [11].

The current levels of coastal urbanization have a significant impact on the landscape. Furthermore, they are considered the main drivers of ecological changes, such as the conversion, degradation, and overexploitation of natural resources, putting coastal areas at risk [12]. The previously mentioned aspects can be observed in the reduction in biodiversity due to anthropic activities [7,13,14] and global losses of wetlands [15,16]. Additionally, the growth levels of coastal urbanization have reached such speeds and complexities that they require a quick landscape planning response. Unfortunately, such planning is unattainable, utilizing traditional methods used by urban planners, such as collecting field data, reviewing historical documents, and previously developed cartographies [17,18].

Remote sensing (RS) offers a sound alternative to traditional methods given its ability to provide a systematic and comprehensive view of the Earth. In this sense, the use of RS has played a significant role, making its technological development indispensable for urban planning, coastal landscape management, and decision-making processes [18,19]. Moreover, using this technique, it has been possible to automate and improve manual mapping, the monitoring of land use transitions and the growth of urban centers and quality of life indicators, population estimates, and evaluate social vulnerabilities, among others [20–23]. In addition, this technique has also contributed to the development of thematic and land use maps, especially in cities, enabling the analysis of changes and trends, and their relationship with urbanization and industrialization processes [24–26] through a series of methods and classification techniques based on pixels and objects. Furthermore, it has also contributed to understanding the types of urban growth and their different underlying issues [27,28]. Finally, advances in the design and development of analysis techniques have allowed their use in researching and managing coastal ecosystems, such as wetlands, dunes, beaches, and estuaries [29].

Despite this, in heterogeneous coastal landscapes and the seasonal variability of some land covers, classification becomes difficult, generating confusion among land covers. Additionally, the presence of wetlands along with urban areas increases the challenge associated with land cover classifications, because they present a similar spectral response to dunes, beaches, and bare soil. Therefore, the heterogeneity of coastal zones and their highly dynamic conditions make them exceptionally difficult to map [30–32]. Despite the difficulty of mapping these areas, numerous authors have contributed various techniques and analyses to improve the classification accuracy and generate reliable maps that can support decision making [33,34]. These techniques focus mainly on pixel, subpixel, and object classification methods. Their contributions range from testing new classifiers (parametric and non-parametric) to incorporating additional information such as vegetation indices, texture metrics, and other statistical analyses, such as principal component analysis (PCA) [35–39].

Additionally, RS is one of the main tools for the delimitation of coastal wetlands, allowing to know their structure and vegetational and hydrological dynamics [40,41]. However, according to Cowardin [42], Ramsar [43], and Tiner [44], to delimit a wetland, it is necessary to respect at least two conditions: (1) water covers the soil; and (2) the presence of hydric or poorly drained soils (3) present temporary flooding at least once a year. Given these conditions and through RS, it is possible to establish some of these two criteria. Alternatively, it can be achieved indirectly, by determining the vegetation cover that is adapted to this environment, also known as hydrophilic vegetation [41,45]. These criteria are also taken by several countries, such as Canada, Australia, New Zealand, The United States, and other European countries [44]. For example, in Chile, the delimitation, protection, and monitorization of wetlands have been drafted in a new law that allows the protection of urban wetlands, based on the criteria mentioned above and proposing

the following: (1) the presence of hydrophilic vegetation; (2) the presence of hydric or undrained soils; and (3) permanent or temporary flooding regime.

For this reason, Chile's coastal wetlands in urban areas need to be delimited and monitored urgently. This study aims at classifying land cover in a coastal system embedded in the metropolitan area of Concepción, Chile, based on a machine learning framework using high-resolution, remotely sensed data. The area is experiencing urban expansion, with intense environmental and political conflicts for the conservation and delimitation of its ecosystems. The specific objectives are: (1) to evaluate the potential of high-resolution imagery to discriminate land covers in a coastal ecosystem using Support Vector Machine (SVM) and Random Forest (RF); (2) to evaluate the impact of including additional data both derived from images and ancillary sources on the performance of the classifiers; and (3) to compare the best results with the delimitations proposed by public institutions in Chile.

## 2. Materials and Methods

### 2.1. Study Area

The Metropolitan Area of Concepción (MAC) is located in the south-central zone of Chile, from 36°44′ S to 37°45′ S and from 73°10′ W to 72°73′ W, in the province of Concepción, which is part of the Biobío Region (Figure 1). It comprises 11 municipalities, having a total population of 1,072,239 in 2017, which makes it the second most populated area in Chile after the Santiago Metropolitan Area [46]. It has an approximate area of 2830 km$^2$ [47].

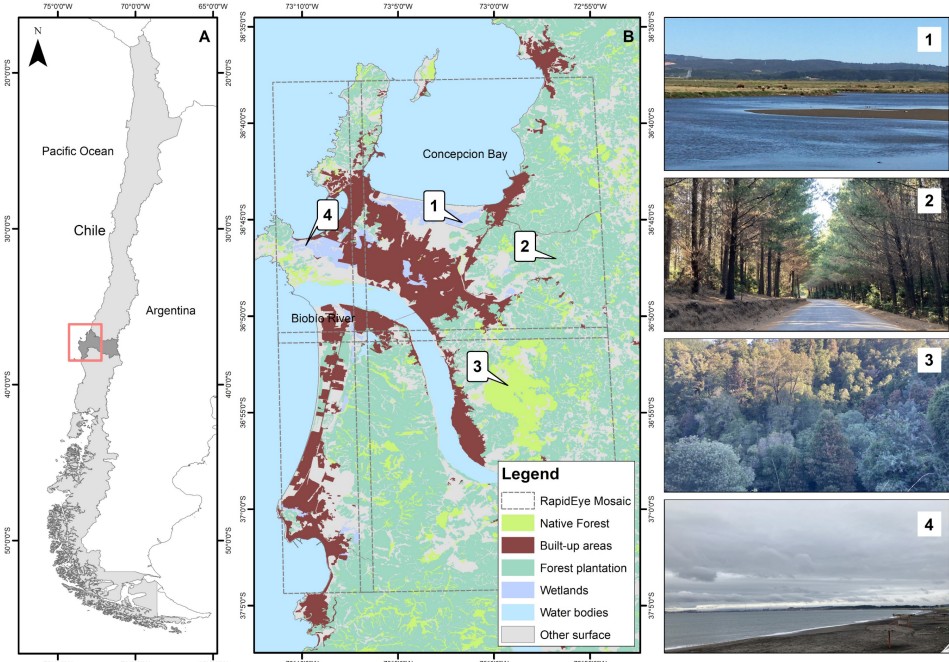

**Figure 1.** Study area Metropolitan Area of Concepción (MAC): (**A**) MAC location in the country; (**B**) Land cover distribution in the MAC; (B1) Estuary of the Andalién River; (B2) Forest plantations; (B3) Native forest in the National Park of Nonguén (Protected Area); (B4) San Vicente Bay between beaches and dunes.

Most of the MAC is in the coastal plains surrounded by the Chilean Coastal Range, which reaches up to 1500 m above sea level. Lacustrine and dune systems are also found [48]. It has a mild climate, with rainfall fluctuating between 1200 and 2000 mm per year [49]. This area was significantly affected by the 2010 Maule earthquake, which disturbed much of the central-south zone of Chile (Mw ≥ 8.5), generating a large number of areas altered by the tsunami and infrastructure damage due to the liquefaction phenomenon along the coast in landfills areas [50,51].

The MAC presents several pressures and urbanization processes characterized by dispersion concentration, drastically altering landscape connectivity [47]. This urban growth has significantly impacted biodiversity, where urban areas have replaced native ecosystems, and wetlands have been destroyed, fragmented, filled, and invaded by exotic species [47,52,53]. Currently, the MAC's growth is expected to continue increasing and using the natural spaces around it, mainly growing on wetlands, dunes, and other natural covers [54].

### 2.2. General Methodology

The methodological steps to accurately classify the coastal landscape of MAC are detailed in Figure 2. First, the necessary radiometric corrections were made to normalize RapidEye's high-resolution images. Second, the information was uploaded to Google Earth Engine (GEE), a platform that made it possible to program the necessary procedures to carry out the analyses. Third, additional information from the spectral bands and a digital terrain model (DTM) were generated to strengthen the final product. In the case of SVM, the optimal parameters of the models were adjusted to improve the classification and accuracy. At the same time, for RF, each band's contribution was calculated to understand the degree of importance in the classification models.

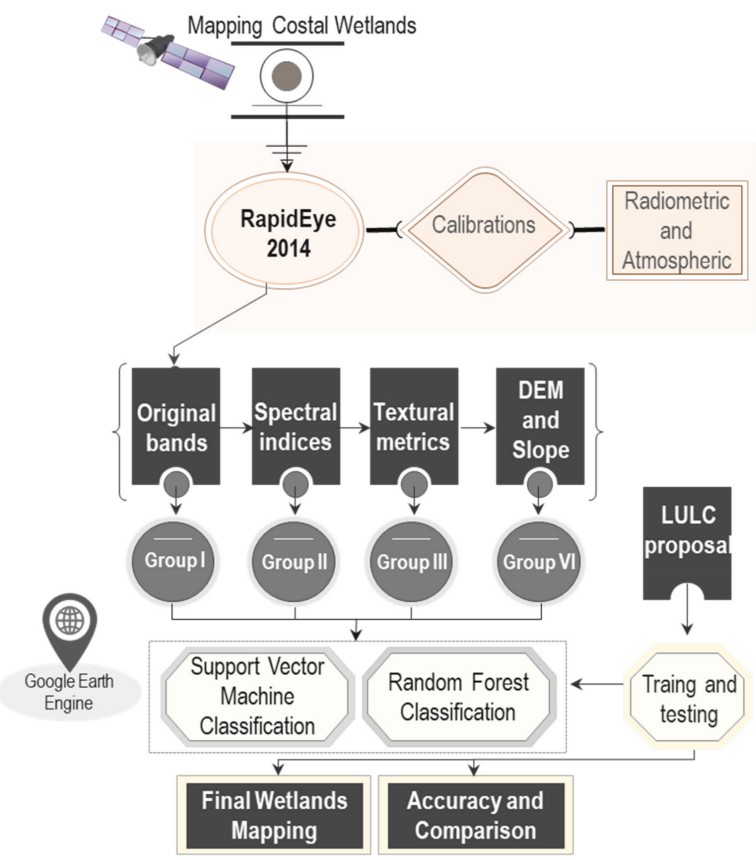

**Figure 2.** Flowchart of the methodological framework developed.

### 2.3. Image Data, Pre-Processing, and Digital Terrain Model Data

Four RapidEye images (http://planet.com, accessed on 7 September 2020) (Table 1) were acquired and mosaicked to cover the entire study area. These images have been widely used in cities, mainly to delimit urban forms of development, identify vegetation in urban environments, detect embedded ecosystems in urban areas, and monitor agricultural uses [30,55–58], due to their high spatial resolution. The images had a level 3A, meaning they incorporate geometric correction and topographic correction based on 30

and 90 m DTM and projected in Universal Transverse Mercator (UTM), Zone 18 South, Datum WGS84.

**Table 1.** Characteristics of the RapidEye images used.

| Characteristics | RapidEye Mosaic |
| --- | --- |
| Pixel Size | 5 m |
| Spectral Resolution | Red-Green-Blue (440–685 nm) Red Edge (690–730 nm) Near Infrared (760–850 nm) |
| Pixel Depth | 16 bit |
| Product Size (column and rows) | 6271 × 9802 |
| Date range images (mm-dd-yyyy) | 3-14-2014 3-15-2014 3-13-2014 3-15-2014 |

The radiometric correction of the images was carried out in two steps. First, absolute radiance was transformed to top of the atmosphere (TOA) reflectance values. Second, an atmospheric correction model was applied based on the dark object subtraction (DOS) method [59].

Additionally, a DTM ALOS PALSAR (http://asf.alaska.edu/, accessed on 13 August 2021) with a 12.5 m spatial resolution was incorporated and used to obtain the slope in sexagesimal degrees. Finally, both layers were resampled at 5 m.

*2.4. Image Processing*

Eight spectral indices were calculated from the original bands to increase the discrimination of the different types of land uses. Table 2 shows the spectral indices derived along with their definition. Each of these indicators were selected to highlight different aspects of the selected coverages. For example, some indicators were used to differentiate between urban and burned areas. Others were intended to differentiate between distinct vegetation types, while the latter helped to improve the discrimination of wetlands.

**Table 2.** Spectral indices.

| Indicator | Formula | Reference |
| --- | --- | --- |
| Burning Area Index (BAI) | $\frac{1}{(0.1-Red)^2 + (0.06-NIR)^2}$ | [60] |
| Enhanced Vegetation Index (EVI) | $2.5 \frac{(NIR-Red)}{(NIR+6\times Red - 7.5\times Blue + 1)}$ | [61] |
| Modified Simple Ratio (MSR) | $\frac{\left(\frac{NIR}{Red}\right)-1}{\left(\sqrt{\frac{NIR}{Red}}\right)+1}$ | [62] |
| Normalized Difference Vegetation Index (NDVI) | $\frac{(NIR-Red)}{(NIR+Red)}$ | [63] |
| Normalized Difference Water Index (NDWI) | $\frac{Green-NIR}{Green+NIR}$ | [64] |
| Non-Linear Index (NLI) | $\frac{NIR^2-Red}{NIR^2+Red}$ | [65] |
| Simple Ratio (SR) | $\frac{NIR}{Red}$ | [66] |
| Transformed Chlorophyll Absorption Reflectance Index (TCARI) | $3\left[(\rho 700 - \rho 670) - 0.2(\rho 700 - \rho 550)\left(\frac{\rho 700}{\rho 670}\right)\right]$ | [67] |

Table 2 shows the spectral indices, where Blue, Green, Red, and NIR represent bands 1 to 5 consecutively of RapidEye, and p corresponds to the wavelength (nm).

Texture metrics from the gray-level co-occurrence matrix (GLCM) were also calculated and applied to improve the discrimination of classes with similar spectral behavior [68,69].

The texture metrics were derived from the first PC obtained from the original bands. Table 3 shows the texture metrics derived [70] with a configuration of 0°–45°–90° in a 9 × 9 window.

**Table 3.** Textural metrics gray-level co-occurrence matrix (GLCM).

| Indicator | Formula | Reference |
|:---:|:---:|:---:|
| Mean | $\sum\limits_{i=1}^{Ng} \sum\limits_{j=1}^{Ng} i * P(i,j)$ | |
| Variance | $\sum\limits_{i=1}^{Ng} \sum\limits_{i=1}^{Ng} (i-\mu)^2 \times P(i,j)$ | |
| Homogeneity | $\sum\limits_{i=1}^{Ng} \sum\limits_{i=1}^{Ng} \frac{1}{1+(i-j)^2} P(i,j)$ | [70] |
| Contrast | $\sum\limits_{i=1}^{Ng} \sum\limits_{j=1}^{Ng} P(i,j)(i-j)^2$ | |
| Entropy | $\sum\limits_{i=1}^{Ng} \sum\limits_{i=1}^{Ng} P(i,j) \, log(P(i,j))$ | |
| Second Moment | $\sum\limits_{i=1}^{Ng} \sum\limits_{j=1}^{Ng} \{P(i,j)\}^2$ | |

Table 3 shows the textural metrics gray-level co-occurrence matrix, where $i$ is the gray intensity value of each pixel, $P(i,j)$ is the gray intensity transition from $i$ to $j$ pixels, and Ng is the number of distinct grey levels in the quantized image.

### 2.5. Training and Testing Data

Since both classifiers that were used are supervised classifiers, a training dataset was required for the algorithm to learn the patterns of the different classes to be discriminated. Thus, a sample of 361 points was collected in the field using a GARMIN Etrex 20x GPS to train the algorithms. This initial sample was augmented through the visual interpretation of the images, using the field collected data as reference, increasing the total sample to 1458 points. Finally, the sample was split into a training dataset (70% of the points) and leaving the remaining 30% of the points as an independent validation set to evaluate the results of the classifications.

### 2.6. Coastal Ecosystems Land Covers within the MAC

A critical aspect when classifying an image is the selection of the categories to be discriminated. In this study, the selection of land covers has been adapted to previous research in the area, emphasizing the detection of coastal ecosystems inserted in urban areas for planning coastal cities. For this purpose, land covers similar to Rojas et al. [47] were proposed for the study area, with eleven categories (Figure 3).

### 2.7. Classification Algorithms and Accuracy Assessment

Two widely used machine learning algorithms were applied to classify the images. The first was the non-parametric statistical learning classifier "Support Vector Machine", initially formulated by Vapnik [71]. The method's objective is to find the optimal hyperplane that separates each set of points according to its class. The optimal hyperplane refers to a decision boundary that reduces the number of misclassified points. The hyperplane consists of a line that assumes that the data are linearly separable in their simplest version. Usually, this condition does not occur in remote sensing applications. It is common to apply the kernel trick [72] to solve this problem, which projects the data into a higher dimensional feature space where the classes may be linearly separated. Multi-class problems, such as classifying land covers, are solved by reducing the problem to a set of binary problems.

Two approaches can be found, "one against one" and "one against all" [73]. For this study, the kernel trick was performed by means of a radial basis function (RBF) kernel, commonly used in remote sensing applications [30,74–76]. The gamma parameter determines the width of the kernel, whereas the cost parameter controls the penalty associated with wrongly classified training samples. Finding the optimal values of these parameters is essential for the performance of the SVM. These optimal values were found using a grid search approach [77].

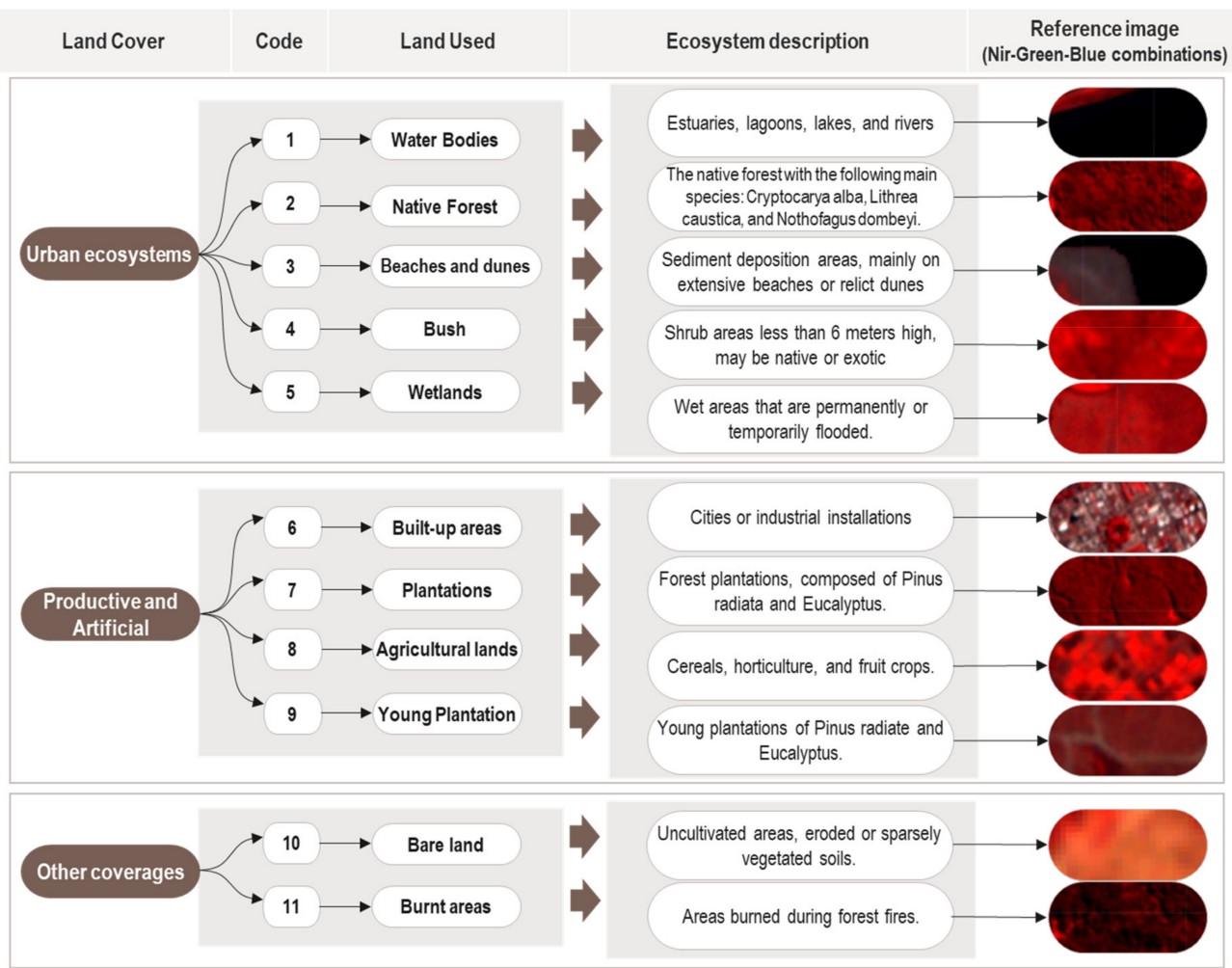

**Figure 3.** Land covers selected for the Metropolitan Area of Concepción, Chile, along with their description.

The second algorithm applied was "Random Forest", proposed by Breiman [78]. It consists of a classifier that uses multiple decision trees (ensemble classifier), each created using a random sample of training points through replacement sampling. Approximately two-thirds of the sample are used for training and one-third for cross-validation. Each decision tree gradually divides the attribute space (feature space), seeking to maximize the separation of classes. A random sample is selected from the set of attributes to generate each tree. The number of decision trees and the size of the subset of attributes (features) are considered hyperparameters. The decision to belong to a class of a pixel is made by voting the decision trees; each tree gives a class label for each pixel, considering the class label that most trees chose as a final result [58,79]. Random Forest has been widely used to classify satellite images [58,80,81]. The hyperparameter tuning was carried out using a randomized cross-validation search, in addition to finding the optimum number of trees

and number of features per split for each classification group. Further, the implementation of the Random Forest algorithm allowed us to obtain the importance of each band for the detection of the proposed land uses, as described in Section 3.2 of the Results.

Eight land cover classifications were applied, four for SVM and RF, respectively, where: (1) RapidEye spectral bands were used alone (Group I); (2) spectral indices (Table 2) were included (Group II); (3) texture metrics were added (Group III); and (4) topographic information was incorporated (Group IV). The accuracy of each classification was evaluated utilizing the metrics derived from the confusion matrix (overall accuracy, omission and commission errors, and the kappa index). Moreover, because the precision relationship between classifiers does not necessarily prove the superiority of one classifier over another [82], the McNemar [83] statistic was applied to the best evaluated classifications of each model (SVM and RF), expressed by the following equation:

$$X^2 = \frac{(f_{12} - f_{21})^2}{f_{12} + f_{21}} \tag{1}$$

In the equation, $f_{12}$ corresponds to the number of cases that are classified incorrectly by classifier 1, but correctly classified by 2, and $f_{21}$ is the number of cases correctly classified by classifier 1, but incorrectly classified by classifier 2.

### 2.8. Comparison with Official Surfaces

To validate the resulting surfaces that gave better accuracy, they were compared with two official boundaries generated by the Ministry of Environment of Chile (MMA). The first one corresponds to the National Inventory of Wetlands, which used similar vegetation criteria and Landsat OLI 8 (2020) images and was calibrated by the Agriculture and Livestock Services of Chile (SAG). The second is the Rocuant-Andalién wetland boundary generated by the Global Environment Facility and the Ministry of Environment of Chile [84] based on multicriteria analysis. This analysis includes biological, geomorphological-geological, hydrological, topographical, vegetational, and intertidal aspects. However, only the vegetation, intertidal, and topography criteria were selected for the comparison.

## 3. Results

### 3.1. Hyperparameter Models and Accuracy

The grid search approach for the SVM parameters yielded the values shown in Table 4. In addition, the results of the hyperparameter search for Random Forest in each of the groups were incorporated.

**Table 4.** SVM-RBF calibration parameters for each parameter.

| | SVM | | RF | |
|---|---|---|---|---|
| **CODE_IMAGE** | **Cost** | **Gamma** | **N° Trees** | **N° Features** |
| I | 3 | 4 | 220 | 3 |
| II | 4 | 1 | 224 | 7 |
| III | 4 | 0.5 | 384 | 11 |
| IV | 4 | 0.5 | 312 | 12 |

Table 5 shows the performance metrics derived from the confusion matrix obtained for each classifier and scenario. In general, the SVM and RF models' performance increased when incorporating additional information, i.e., indices, texture, and elevation data, except for the SVM group IV. Thus, the overall accuracy of the SVM classifier began with 82.03%, increasing by 0.14% when incorporating the spectral indices (group II). A higher increase was observed with the incorporation of the texture metrics (group III), augmenting the overall accuracy by 11.25% compared to group I. Finally, when incorporating the topographic information (group IV), the accuracy decreased imperceptibly to 92.73% (−0.55% compared to the previous group). A similar trend was observed for RF, where the performance was

increased in all groups, starting with an overall accuracy of 86.63% for the original bands (group I), which later increased by 0.13% for group II and by 4.25% for group III. However, unlike SVM, RF continued to increase when incorporating the topographic information (group IV), reaching a final accuracy of 95.88%, that is, 3.15% higher than SVM. The kappa indices showed similar behavior, increasing from 0.80 to 0.92 from group I to group IV of SVM, while for RF, it increased from 0.86 to 0.95.

**Table 5.** Accuracies obtained for each classifier and different variable combinations.

| Variable Combination | | SVM | | | | RF | | | |
|---|---|---|---|---|---|---|---|---|---|
| | | I | II | III | IV | I | II | III | IV |
| Water bodies | User's | 95.45 | 95.4 | 98.90 | 98.89 | 96.55 | 96.59 | 98.88 | 100.00 |
| | Producer's | 93.33 | 92.22 | 100.00 | 98.89 | 93.33 | 94.44 | 97.78 | 100.00 |
| Native forest | User's | 80.56 | 75.56 | 93.58 | 91.96 | 80.7 | 81.82 | 87.61 | 97.20 |
| | Producer's | 79.09 | 92.73 | 92.73 | 93.64 | 83.64 | 81.82 | 90.00 | 94.55 |
| Dunes | User's | 58.06 | 68.57 | 84.51 | 88.57 | 67.09 | 66.67 | 83.78 | 94.37 |
| | Producer's | 46.75 | 31.17 | 77.92 | 80.52 | 68.83 | 62.34 | 80.52 | 87.01 |
| Shrub | User's | 89.55 | 92.54 | 97.10 | 98.53 | 95.39 | 96.88 | 94.12 | 97.18 |
| | Producer's | 84.51 | 87.32 | 94.37 | 94.37 | 87.32 | 87.32 | 90.14 | 97.18 |
| Wetlands | User's | 91.01 | 83.9 | 96.12 | 94.27 | 93.89 | 93.16 | 93.26 | 99.61 |
| | Producer's | 94.92 | 95.7 | 96.88 | 96.48 | 96.09 | 95.7 | 97.27 | 99.22 |
| Built-up areas | User's | 75.71 | 76.28 | 94.31 | 92.66 | 86.46 | 88.02 | 92.45 | 94.52 |
| | Producer's | 72.6 | 74.89 | 90.87 | 92.24 | 75.8 | 77.17 | 89.50 | 94.52 |
| Bare soil | User's | 76.1 | 75.84 | 88.28 | 88.28 | 80.08 | 82.28 | 85.96 | 90.91 |
| | Producer's | 86.04 | 91.89 | 95.05 | 95.05 | 92.34 | 94.14 | 90.99 | 94.59 |
| Overall accuracy | | 82.03 | 82.17 | 93.28 | 92.73 | 86.63 | 86.76 | 90.88 | 95.88 |
| Kappa | | 0.8 | 0.8 | 0.92 | 0.92 | 0.86 | 0.85 | 0.90 | 0.95 |

Regarding user's and producer's accuracies (Table 5), the performance of some land uses was highlighted, particularly the dunes, which in the first group of SVM reached 47% precision, increasing up to ~80% for group IV. The water bodies were those with the highest accuracy, reaching 98.89% and 100% (user's) for SVM and RF, respectively, while the worst accuracy recorded for SVM-I are the dunes with 58.06% for user's and 46.75% for producer's, possibly due to their low spectral separability. In contrast, for RF, it was found in group II with 66.67% for user's and 62.34% for producer's. Finally, wetlands showed notable improvements for both user's and producer´s accuracy when incorporating the DTM band, reaching values >90%.

The increase in overall accuracy (Table 5) was visually observed as an increased spatial coherence of the classifications. For SVM (Figure 4), a decrease in the "salt and pepper" effect and greater generalization are seen. For the most part, the incorporation of information helped to eliminate heterogeneity within the same land cover classes, which can be seen in the first five images of Figure 3, where the water body, wetlands, urbanization, scrubland, and young plantation use showed important improvements in surface delineation. Likewise, a substantial improvement in the delimitation of the native forest located in Nonguén Park can be observed in the last five images of Figure 3, and the city's shape and burnt areas.

A similar pattern was observed with RF (Figure 5), where the first variable combinations (see images numbers 2 and 7) show the "salt and pepper effect". However, as the model incorporates information, the classification is improved. The previously mentioned aspects can be seen in images number 5 and 10 in Figure 4, which show the improvement in the discrimination of the urban, native forest, and wetlands. In addition, dunes and beaches, bare land, and water bodies eliminate noise and improve the delimitation of their borders. Burnt areas also showed better delimitation. However, some noise is observed

for this land cover near the water bodies and dunes and beaches. There is also significant noise from the native forest in some sectors with forestry plantations.

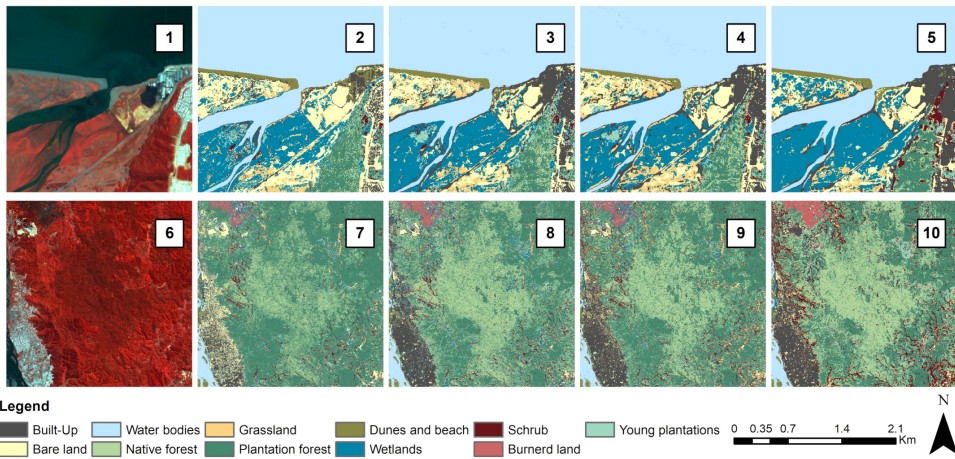

**Figure 4.** Coastal ecosystem land cover comparison with SVM-RBF. From 1 to 5 is possible to observe the Andalién estuary and the marsh, while from 6 to 10 the Nonguén park surrounded by urbanization with SVM-RBF.

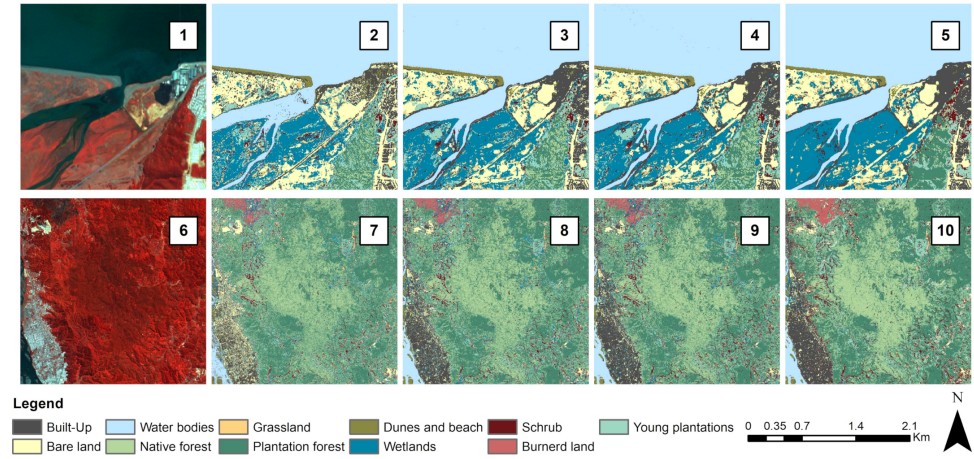

**Figure 5.** Coastal ecosystems cover comparison with RF. From 1 to 5 is possible to observe the Andalién estuary and the marsh, while from 6 to 10 the Nonguén park surrounded by urbanization with RF.

### 3.2. Random Forest Variable Importance and Statistical Results

The importance was calculated in coastal ecosystems (Bodies of water—Native forest—Beaches and dunes—Shrub—Wetlands) and built-up areas, on RF-IV (Figure 6), which showed the best performance (around 95.88% accuracy) and underwent a thorough visual inspection, showing fewer errors than the other models (Table 5).

Importance was concentrated in the spectral indices and with a higher proportion for the ancillary DTM data, while the incorporation of texture metrics did not make significant contributions. For the urban class, it was observed that the most significant importance was contributed by the bands NDWI (0.19), BLUE (0.18), NDVI (0.14), and MSR (0.11). Notably, the texture information (see Table 3) had greater relevance for urban use regarding other uses, having values on average of 0.09. In the case of the water bodies land cover, the most influential bands were: NIR (0.26), NDWI (0.19), NLI (0.16), DTM (0.15), and NIR edge (0.13). Meanwhile, for native forest land cover, the most influential bands were the DEM (0.45) and RED bands (0.21). The key bands for dunes land cover were DTM (0.32)

and NDWI (0.28). For wetlands, the DTM again reached a maximum value (0.42), which is the most important band for the delimitation of these ecosystems. Finally, in the case of scrubland, the most crucial bands were the MSR (0.2), SR (0.2), NDVI (0.2), and DTM (0.2) bands.

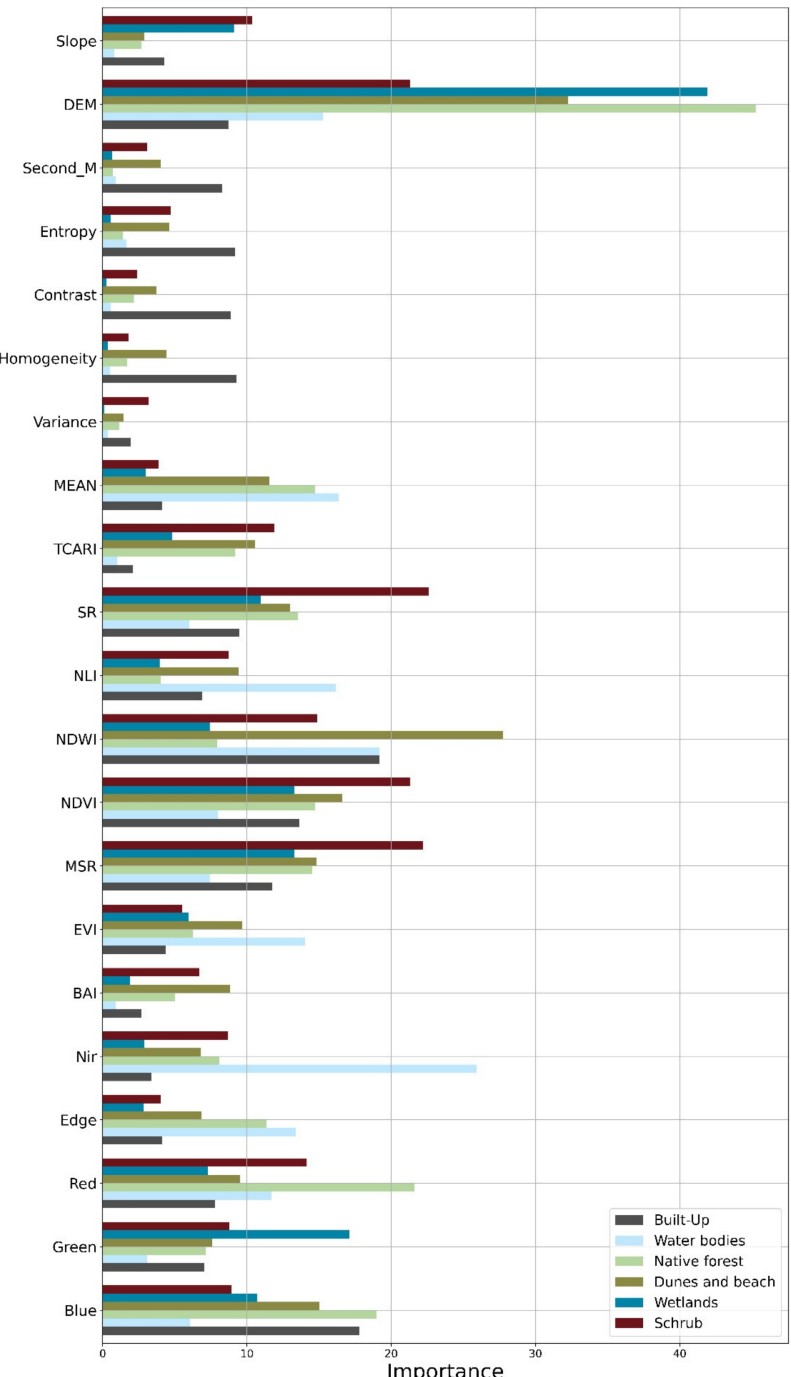

**Figure 6.** Importance of each band for the RF classifier and group IV.

The statistical results show that RF-IV versus SVM-RBF III yielded a chi-square of 32.29 with a *p*-value <0.001. Therefore, they have significant differences. The aforementioned indicates that RF-IV has statistically significant differences in SVM accuracy, and consequently provides a classification with fewer errors.

### 3.3. Comparison of Results with Official Limits

Once the best classifiers of each group were determined, the overlap between the boundaries of coastal wetlands was assessed (Figure 7). RF-IV gave an area of 860 ha, which corresponds to the smallest area detected through the RapidEye image. It is possible to find fragments of wetland detection that were not cataloged by the boundaries of NI (2020) and GEF-MMA (2021). SVM-III detected an area of 1084 ha. Although it is similar to RF-IV due to having the same criteria, it is possible to observe that it detected wetlands in areas close to the dune body and confusing parts of the urban area. Although the official boundaries differ by 349 ha, both contain a large part of the area of wetlands detected by the study.

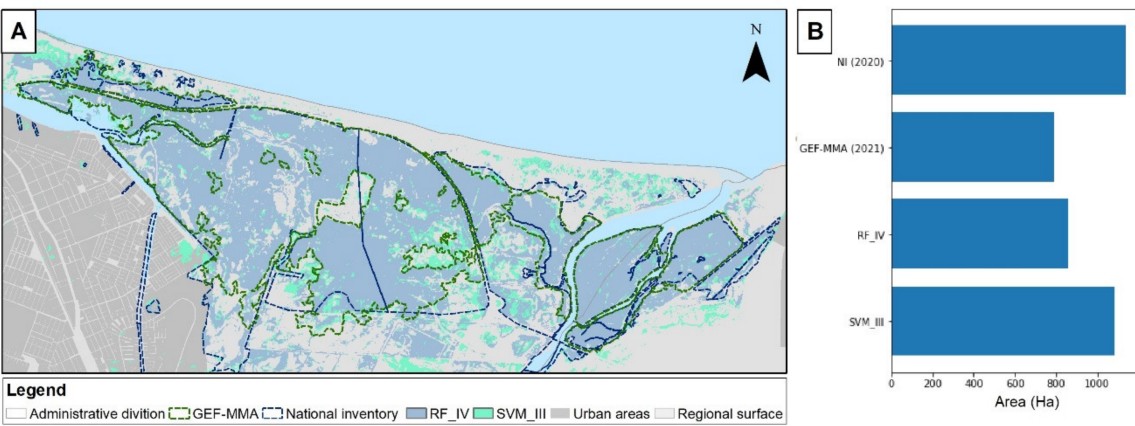

**Figure 7.** Comparison of the classification results with the official limits. In (**A**) it is possible to observe the distribution of the classifiers and the official limits, in (**B**) the difference of the calculated surfaces can be observed.

## 4. Discussion

### 4.1. Model Uncertainties

During the last two decades, a wide variety of machine learning algorithms have been proposed to study land-use coverage, with RF and SVM showing the best results to date, surpassed only by neural networks [81]. Although its implementation should not be considered an important advance in the development of classifiers, due to the large number of existing studies, this study sought to highlight its efficiency and versatility to improve inventories and the monitoring of coastal wetlands with the incorporation of spectral, textural, and topographical variables [38,85,86]. One of the main characteristics of these classifiers is their ability to handle multidimensional data, deal with the non-linearity of the variables, show tolerance to unbalanced samples, and reduce processing and calculation time [87–89].

Regarding these characteristics of the classifiers, this study is not an exception. Both classifier methods obtained similar accuracies and patterns when incorporating additional information, with similar results to other studies [30], obtaining an overall accuracy of 93.07% when using RapidEye images and classifiers SVM and RF in the coastal landscapes in the MAC. SVM showed promising results, with an average overall accuracy of 88% in all models used. It showed improvement when increasing information in at least three scenarios (I–III), especially when incorporating the texture metrics, which have also been reported by other authors [90]. Meanwhile, incorporating topographic information did not increase accuracy in the classification, but a negligible decrease.

RF showed a more stable behavior, reaching an average overall accuracy of 90%. According to the literature, it is a model built for multiple weak learners, decreasing variance. Additionally, it is resistant to noise and overfitting [81]. In this study, RF effectively incorporated the DTM information, which had the most significant importance in terms of

accuracy for the majority of the classes. Therefore, both classifiers yielded results with a few differences, with SVM being superior in group III.

Considering all groups generated, the training areas used in this study allowed to obtain results with over 90% accuracy. However, the literature has shown that it is ideal to have a large number of samples for RF, as this is directly related to accuracy [91], while SVM has shown to be efficient with limited numbers of samples [92]. Nevertheless, SVM is particularly sensitive to the calibration of the hyperparameters (Gamma and Cost), which is a critical factor when classifying [92,93].

In terms of land cover, the main challenges found for SVM-I and II were among the urban-bare, urban-sediment, plantation-native, burnt-bare, wetlands-agriculture, and wetlands-young plantations and uses. These confusion issues were reduced in SVM-III for urban-bare, urban-sediment, and bare-sediment. However, for SVM-IV, the errors from I and II returned, reducing accuracy. Meanwhile, the main errors for RF I and II were urban-bare, urban-sediment, and native-plantations. By incorporating texture and the elevation model in RF III and IV, they were reduced to bare-sediment and plantations-young plantations. These results coincide with some studies where the use of spectral and texture information has led to obtaining precise ground use classifications [38,94].

### 4.2. Comparison with Chilean Cases

Chile has a great diversity of wetlands, is highly productive, and is considered a vital biodiversity hotspot due to their high endemism [95–97]. However, authors agree that they are strongly subjected to the pressure of human activities, reducing their extension, connectivity, and biodiversity [95,98]. The MAC area is no exception to this trend since, if we compare the results of the classifications on a total of 9253 ha with the latest studies carried out with Sentinel-2 using the RF classifier [99], we can establish that in the period of 2014–2019, there was a 494 ha increase in urbanization, while wetlands lost were 1696 ha. This loss trend in wetlands in the coastal zones was already described by Pauchard [52] and Rojas et al. [47,54].

To reverse this trend, Chile subscribes to the Ramsar convention allowing the protection of 16 wetlands (363,927 ha). Most of these wetlands are delimited based on biological criteria identifying hydrophilic vegetation and aquatic biota [100]. Additionally, the promulgation of the Urban Wetlands Law allows the protection of wetlands totally or partially inserted in urban limits, thus generating the need to start delimiting ecosystems. The previously mentioned has led the Ministry of the Environment of Chile to elaborate the National Wetlands Inventory, which was elaborated using the re-classification of spectral indicators, specifically of NDVI [101], which led to a series of errors of interpretation and omission of other areas with temporary flooding [102]. However, this was corrected with a new guide for the delimitation of wetlands [101], which suggests identifying the surface using RS that must be complemented with other information from scientific knowledge, such as hydrology and fieldwork.

These differences in delimitation methods could explain the results that we obtained through the comparison with the surfaces proposed by official institutions. These showed differences possibly due to the different classification methods, the vegetation criteria, and the differences in spatial and spectral resolution scales of the images (RapidEye vs. Landsat 8 OLI) [41]. However, the costs of the images must be considered mainly in the context of the limited access to resources [103].

### 4.3. Coastal Wetland Worldwide Implications

These classifiers have been widely discussed due to their success and application in various types of landscapes [81]. Moreover, their implementation in heterogeneous coastal landscapes has been successful, and their application has also served as a validation method for new classification methods, such as deep learning [30,104,105].

Regarding the RS data utilized in this study, RapidEye images were widely used because of their high spatial resolution that, together with their red edge band, allowed

to obtain precise classifications and achieve users' objectives. This is particularly true in heterogeneous landscapes or those dominated by agriculture, urban areas. and wetlands [106–109]. In some cases, these images are critical for differentiating some of the target classes, especially in classes of small areas, such as wetlands areas and dunes. There are also other more advanced classification methods, such as deep learning and object-based classification. These classification methods have proven to be superior to the traditional ones (SVM and RF), but they require greater user knowledge in RS in addition to lacking expertise in public institutions, which could be a weakness in developing countries [110,111].

According to the results obtained, these are practical limits and represent a first approximation for decision makers, with surfaces close to those proposed by official limits. However, as discussed above, these results must be improved by incorporating studies on the periodicity of floods, the inclusion of radars, indicators of hydric soils, and the recognition of the composition of plant communities [112].

## 5. Conclusions

This study showed the ability of high spatial resolution satellite imagery to map land covers in a coastal ecosystem (urban wetlands) of urbanized areas, characterized by its high dynamics and tsunamis and flood risk. SVM and RF, two machine learning algorithms, proved their capability to accurately discriminate land covers when applied to high-resolution imagery. The case of RF provides insight into whether the variables can aid in improving these classifications. It also showed that, by incorporating spectral and texture information and topographic information, all classifiers improved their performance, except for the SVM group IV, which reported a negligible decrease in accuracy when using topographic information, with texture being more helpful. The main findings indicate that RF performed better than SVM when all the information was applied, with DTM being relevant for detecting coastal ecosystems. However, the SVM results should not be disregarded, as once textural information was incorporated, this classifier obtained results with 93% accuracy, only 2% lower than the best result achieved by RF.

These results contribute to detecting the land covers in coastal areas pressured by urban growth using tools, such as high-resolution satellite images. The methods allowed differences in similar land cover classes as wetlands, sediments areas, beaches, and between urban areas and bare lands with precise accuracy. However, it is always necessary to carefully manage the confusion between urban-bare, urban-sediment, and bare-sediment. In this regard, the calculation of vegetation indexes, texture metrics, and additional information improves the processing of classifications of coastal ecosystems.

In addition to their low cost and easy access data, RapidEye images proved to have enough spatial and spectral resolution to successfully differentiate the different covers that were considered. Likewise, both classifiers (SVM-III and RF-IV) showed to be accurate and could be used in decision-making and urban planning processes, especially in cities that lack governance. Furthermore, these techniques decrease the amount of post-classification work and facilitate the supervision of cities' growth in the habitat fragmentation and ecological connectivity of coastal ecosystems.

Finally, the processes applied in this work can be easily replicated for other coastal wetlands, since the platform in which they were applied is freely available, in addition to generating reliable delimitations, allowing decision makers to accelerate the delimitation process.

**Author Contributions:** Conceptualization, J.M., M.G. and C.R.; methodology, J.M., M.G., F.U., V.N., O.R. and C.R.; software, J.M. and F.U.; validation, J.M. and F.U.; formal analysis, J.M., M.G., F.U. and C.R.; investigation, J.M., M.G., F.U. and C.R.; resources, C.R.; writing—original draft preparation, J.M., M.G., F.U., V.N., O.R. and C.R.; writing—review and editing, J.M., M.G., F.U., O.R., V.N. and C.R.; visualization, J.M., M.G., F.U., V.N., O.R. and C.R.; supervision, J.M., M.G., V.N., O.R. and C.R. All authors have read and agreed to the published version of the manuscript.

**Funding:** This research was funded by the FONDECYT-ANID program (1190251) and National Doctorate Scholarship 2020 (Grant N°21201439) of the Government of Chile.

**Institutional Review Board Statement:** Not applicable.

**Informed Consent Statement:** Not applicable.

**Data Availability Statement:** Not applicable.

**Acknowledgments:** Juan Munizaga is grateful for the National Agency for Research and Development of Chilean Government ANID, National Doctorate Scholarship 2020, Grant N° 21201439, which supports his PhD studies, and the FONDECYT 1190251, which funded the field work, English edition, and publication costs.

**Conflicts of Interest:** The authors declare that they have no conflict of interest.

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
