# Peer review of "Mapping Coastal Wetlands Using Satellite Imagery and Machine Learning in a Highly Urbanized Landscape"

_sustainability, doi:10.3390/su14095700_

Round 1
Reviewer 1 Report
Overall, I think the authors have presented a complete and valuable research, but their contributions need to be further elaborated in the summary and conclusion.
Firstly, SVM and RF, these two machine learning methods, have been common methods in remote sensing image recognition. Their application and results comparison in this paper cannot be regarded as a very prominent discovery.
Secondly, in fact, the difficulties in the interpretation of remote sensing images around the coastline have been mentioned in the paper, but there is a lack of elaboration on the progress of this research on these difficulties in the abstract and conclusion. It is suggested that the authors should supplement them to highlight the contribution of this paper.
Author Response
The manuscript has been revised and aspects suggested by the other reviewers have been improved. References and phrases have been incorporated to improve the understanding and impact of the article. We appreciate your willingness to read and approve our work.
Please review the attached document for the detail of corrections.

Reviewer 2 Report
This review is on the article “Mapping coastal wetlands using satellite imagery and machine learning in a highly urbanized landscape”
I found this article a very informative, timely requirement.
Author Response
The manuscript has been revised and aspects suggested by the other reviewers have been improved. References and phrases have been incorporated to improve the understanding and impact of the article. We appreciate your willingness to read and approve our work.
Please review the attached document for the detail of corrections

Reviewer 3 Report
Dear authors, you are presenting a very good manuscript that can be improved with minor corrections I copy you in the attached PDF.
All the best.

Author Response
The manuscript has been revised and aspects suggested by the other reviewers have been improved. References and phrases have been incorporated to improve the understanding and impact of the article. We appreciate your willingness to read and approve our work.
Please review the attached document for the detail of corrections

This manuscript is a resubmission of an earlier submission. The following is a list of the peer review reports and author responses from that submission.
Round 1
Reviewer 1 Report
1) General comments
This study uses two machine learning methods (that is, support vector machine and random forest) to finely classify the land cover of coastal ecosystems in Metropolitan Area of Concepcion of Chile based on high-resolution RapidEye images. Both support vector machine and random forest methods have obtained good classification results, validating the capability of high-resolution RapidEye images to monitor and map coastal ecosystems.
Generally, the topic of the investigation is significant, the used technique is acceptable. However, the manuscript is similar to the research work done by Adam et al., making it lack of innovation. With significant reorganization and revision, this study has the potential to be an interesting contribution.
The manuscript lacks a clear hypothesis, thus resulting a very descriptive text. The authors should formulate a valid scientific hypothesis, concentrate on a general scientific framework around this new hypothesis. That could be a useful starting point for a good Introduction part.
As the result I have some major reservations about the contribution of this work to the existing body of knowledge and this lead me to recommend a rejection to this paper.
To wit:
First of all, the objective of this study is not clear. It focuses on the description of support vector machine and random forest algorithms, but in the end, the complete land cover distribution map of the study area was not obtained.
Second, this study only discusses the comparison between the two machine learning methods of support vector machine and random forest, but does not discuss the comparison with other non-machine learning methods.
Third, this study is similar to the research work done by Adam et al., making it lack of innovation.
Additionally, some of the manuscript's expression is not appropriate, and some of the statements in introduction part have no corresponding literature support. Literature research on this statement is needed to support the statement. I will give a detailed explanation in the "specific comments" below, please refer to it carefully and modify it.
Furthermore, the writing of the manuscript requires a lot of English editing, and there are many errors in grammar, spelling and sentence structure. I have provided some suggested changes in the "specific comments" below to improve this situation.
2) Specific comments
Throughout the manuscript:
- All abbreviations in each part of the manuscript including abstract and contents, as well as in the Table and Figure, should be introduced for the first time despite how common or not the abbreviation is.
Abstract
- There is no need to go into details in an abstract. Please pay more attention to the problems solved and the key findings obtained, and a modification is need.
Introduction
- No scientific and reasonable hypotheses are made, making the text too descriptive. The hypothesis should be established and tested through observations, and the hypothesis should be demonstrated point-to-point around the research purpose, and finally, the problems and significance of the research should be summarized.
- The manuscript focuses on the importance of coastal ecosystems in the introduction section, but does not describe the methods used by the predecessors to classify the land cover of coastal ecosystems. I suggest that the author focuses on the following two issues:
(1) Why do you use high-resolution images to monitor coastal ecosystems?
(2) Why do you use machine learning methods to classify land cover in coastal ecosystems?
Materials and Methods
- Figure 1: I suggest that the author add a scale to Figure 1.
- Figure 2: The logic of the flowchart in Figure 2 is not clear, I suggest that the author redraw it.
- Table 1: I suggest that the authors use three-line tables for all tables in the full text.
- Line 206: In this study, only 4 classifications are used, not 8 classifications, but these 4 classifications are the same in support vector machines and random forest algorithms. Please check and correct them carefully.
- Line 214: The formula used in this study is not clear, please check and correct it carefully.
Results
- Table 5: Please explain clearly what C in Table 5 means.
- Table 6: In this study, Table 4 shows that there are 11 types of land use in total, while Table 6 shows how there are only 7 types of land use. Please explain clearly.
- Figure 3: I suggest that the author mark in Figure 1 the specific locations of the two locations used in Figure 3.
- Figure 4: I suggest that the author mark in Figure 1 the specific locations of the two locations used in Figure 4.
- I suggest that the author apply the support vector machine and the random forest algorithms to the study area to obtain the land cover distribution map of the entire study area, and compare the distribution maps obtained by the two methods.
- I suggest that the author delete the content in section “3.2. Random Forest variable importance”, because I don't know what problem the author is trying to solve in this part, and what is the meaning.
Discussion
- I suggest that the author elaborate in the discussion section that adding additional topological information to SVM-III will reduce the classification accuracy of the support vector machine.
- In Table 5, the cost and Gamma parameters of group III and IV are the same. Why does it appear that the accuracy of support vector machines in group III is higher than that in group IV. Please explain clearly.
Conclusions
- I suggest that the author list several key findings of this research in the conclusion section. Please author reorganize the relevant content of the conclusion part.
Reviewer 2 Report
Mapping and identifying areas with higher ecological status (as coastal zone are) are in the interest of environmentalists for a long time to preserve the good condition of the landscape. Information on the occurrence of different types of LC is essential for ecosystem quality evaluation and is involved in many differently designed databases that obtain this information in different ways.
Overall, the present article is well-established and brings a valuable contribution to research in automatic land cover detection in coastal areas. The study's logic was easy to follow, and the structure is already part of the standard and routine identification procedure that applied an automatic machine learning approach. This consists of several steps that are clearly described in the manuscript (MS).
Using remote sensing (RS) in coastal (or deltas, wetlands, and freshwaters) ecosystem application increasing rapidly since 2000. Improvements in sensor technologies, higher spectral and spatial resolution, accessibility of satellite products, and the development of semiautomatic classification methods adopted in commercial/non-comercial software led to a rapid increase in the number of articles apllying these principles of LC classification. Within a few years, many articles were published that use a simplified scheme from satellite data, training/testing, application of machine learning classifier to LC map final product.
In this context, I propose several changes and modifications that will not reduce the quality of the manuscript but will lead to greater added value and contribution for the scientific community. This manuscript can bring this value, not only routine description and comparison of two different classifiers for LC determination.
The manuscript contains quite an extensive literature review in the introduction section, but I would recommend focusing more on RS and application in automatic LC detection in coastal areas. Partially this is incorporated in lines 69-81. I think that this part should be expanded with more detail on the spectral/temporal satellite data properties, classifier selection, and classification process differences. Several studies worldwide used similar methodological procedures to identify LC (more or less similar in spectral or physiognomic chracteristics) in coastal areas and freshwater aquatic ecosystems. This comment is also supported by the fact that only one article cited in the discussion section is mentioned in the introduction. In this context, the work provides surprisingly few arguments about the reasons and benefits of this approach over other methods.
Section 2. Materials and Methods have a logical structure; however, I suggest excluding the study area to a separate chapter. In the image processing (sub-chapter 2.3) missing information about of TOA transformation process (L137). The sub-chapter 2.5 TAT data need more additional information. The classification result depends on the correct training data definition, which is necessary for suitable class separation. In the MS, please define training area statistics. How are the individual classes distributed in TAT data (area or pixel count)? The text is missing information for selecting just these combinations of vegetation indices (8) and texture metrics (6) for this study and their assumed effect on class separation or statistical description of class separation according to the similarity between them. In the methodology section, the authors describe the application of McNamara test statistics. Still, this metric was not used in the result section for classifier evaluation (or I didn’t notice this application). Random Forest variable importance definition and computation were missing in the Methodology section. The way how this metric was calculated is crucial for this data interpretation.
The Discussion section has a small impact, and I recommend its lead in two ways. The first one, as a sub-section dealing with coastal system identification in terms of environmental quality, and the second one emphasizes RS, spectral differentiation, class separation, and ML classifiers comparison.
minor comments:
L135 – pixel size unit change to “m” or “meters” instead of mts
L153 - figure captions should be self-explanatory (the same table 2) and add explanations of equations, what is important to its readability without link reference
L205- wrong sentence (section 4.3 missing in MS)
L214 – error in equation
L256 and 267 - figure captions should be self-explanatory; please describe all numbered figures in Figures 3 and 4.
Reviewer 3 Report
This article applies Support Vector Machine and Random Forests for coastal zone classification. Four scenarios were evaluated, including the original spectral bands, incorporating spectral indices, adding texture metrics, and using digital terrain model and slope. Few comments before publication are provided below.
- For the whole text, do not use abbreviations since you would not use it throughout the text, for example Principal component analysis, used once, so there is no need to write (PCA).
- The first paragraph of the methodology, the authors stated the workflow as first, then second and then stopped. Complete stating the workflow steps or refer to the figure and details are explained in the following sections.
- Figures are poorly presented.
- Figure 1: the numbers are hardy readable and use more contrast colors for land covers.
- Figure 2: polygons are not complete.
- Figure 4: text is not clear.
- Table 1: mts?
- Table 2: add a column explaining the usage of each index.
- Line 150: what is PC?